# Action of the Photochrome Glyght on GABAergic Synaptic Transmission in Mouse Brain Slices

**DOI:** 10.3390/ijms231810553

**Published:** 2022-09-12

**Authors:** Elena Petukhova, Daria Ponomareva, Karin Rustler, Burkhard Koenig, Piotr Bregestovski

**Affiliations:** 1Institute of Neurosciences, Kazan State Medical University, 420111 Kazan, Russia; 2OpenLab Gene and Cell Technologies, Institute of Fundamental Medicine and Biology, Kazan (Volga Region) Federal University, 420008 Kazan, Russia; 3Institut de Neurosciences des Systèmes, Aix-Marseille University, INSERM, INS, 13005 Marseille, France; 4Department of Normal Physiology, Kazan State Medical University, 420111 Kazan, Russia; 5Faculty of Chemistry and Pharmacy, University of Regensburg, 93053 Regensburg, Germany

**Keywords:** Glyght, photopharmacology, azobenzene, glycine receptors, GABA_A_ receptors

## Abstract

Glyght is a new photochromic compound described as an effective modulator of glycine receptors at heterologous expression, in brain slices and in zebrafish larvae. Glyght also caused weak inhibition of GABA_A_-mediated currents in a cell line expressing α1/β2/γ2 GABA_A_ receptors. However, the effects of Glyght on GABAergic transmission in the brain have not been analysed, which does not allow a sufficiently comprehensive assessment of the effects of the compound on the nervous system. Therefore, in this study using whole-cell patch-clamp recording, we analysed the Glyght (100 µM) action on evoked GABAergic inhibitory postsynaptic currents (eIPSCs) in mice hippocampal slices. Two populations of cells were found: the first responded by reducing the GABAergic eIPSCs’ amplitude, whereas the second showed no sensitivity to the compound. Glyght did not affect the ionic currents’ amplitude induced by GABA application, suggesting the absence of action on postsynaptic GABA receptors. Additionally, Glyght had no impact on the paired-pulse modulation of GABAergic eIPSCs, indicating that Glyght does not modulate the neurotransmitter release mechanisms. In the presence of strychnine, an antagonist of glycine receptors, the Glyght effect on GABAergic synaptic transmission was absent. Our results suggest that Glyght can modulate GABAergic synaptic transmission via action on extrasynaptic glycine receptors.

## 1. Introduction

Photopharmacology is a relatively new, but rapidly emerging field of research aiming to develop light controlled chemical compounds for the specific modulation of ionic channel functions, receptors and other cellular components. Photopharmalogical compounds are coupled to photoswitches, such as azobenzene, spiropyran or diarylethene, and have two states (“on” and “off”) that can be toggled photochemically by isomerization or bond formation [1,2,3].

Recently we developed several photochromes capable to modulate in the light-dependent manner the function of GABA and glycine receptors. These receptors are ligand-gated anion-selective channels that allow for chloride ions to pass through the pore and provide the main inhibitory effect across different brain regions, spinal cord and other parts of the nervous system [4,5,6]. When developing such compounds, we paid attention to 7-aminonitrazepam and synthesized a number of derivatives, two of them proved highly promising: AzoNZ1 and Glyght [7,8]. Both carry an azobenzene unit that allows to switch compounds between *cis* and *trans* configurations using different wavelength of illumination. The switching of compounds from the *trans* to the *cis* isomer is induced by ultraviolet (UV) irradiation. The reverse isomerization can be triggered by irradiation with visible or blue light [7,9].

The compounds were originally intended to interact with the benzodiazepine binding site of GABA_A_ receptor and cause its potentiation. However, photochrome constructs exhibited different features. It has been shown that Azo-NZ1 operates as a light controlled blocker of GABA_A_ α1/β2/γ2, GABA_C_ rho2 receptors and also α2 glycine receptors (GlyRs) [7,8]. In *trans*-configuration, Azo-NZ1 caused inhibition of ion channels, whereas in the *cis*-configuration the compound released the inhibition. This was confirmed by several methods: molecular docking calculations, electrophysiological experiments on a cell line with recombinant gene expression and on mouse brain slices [7,8].

The other compound, Glyght (Figure 1a), showed its main effect on glycine receptors. In *trans*-configuration, Glyght caused a weak inhibition of glycine-induced currents, whereas in the UV-induced *cis*-configuration, its inhibitory effect strongly increased [9]. The molecular docking calculations demonstrated that Glyght is a negative allosteric modulator of glycine receptors. In *cis*-configuration it stabilizes the closed state of the ion channel, whereas in the *trans*-configuration it can stabilize the open state [9,10]. Effectiveness of Glyght as a tool for light-controlled modulator the activity of glycine receptors was proven both at heterologous expression and in brain slices [9]. However, action of this photochrome on synaptic GABAergic transmission has not been analyzed. Therefore, the aim of the present study was to analyze the effect of Glyght on GABAergic synaptic currents in brain slices. Our observations demonstrate that the photochrome did not affect the activity of postsynaptic GABA_A_ receptors in hippocampal neurons but showed small inhibition of stimulated GABAergic synaptic currents. This effect is presumably carried out through the action on extrasynaptically expressed glycine receptors, which provide a tonic inhibitory modulation of synaptic transmission.

## 2. Results

### 2.1. Glyght Action on GABAergic Evoked Inhibitory Postsynaptic Currents

Paired, with the interval of 200 ms, GABAergic evoked Inhibitory Postsynaptic Currents (eIPSCs) were recorded from the mouse (P9–P15) pyramidal cells of the hippocampal CA1 area using the whole-cell configuration of the patch-clamp technique at a holding potential of −70 mV in the presence of CNQX 10 µM and D-AP5 40 µM. To excite GABAergic interneurons, the bipolar stimulating electrode was placed in the *stratum radiatum* in close proximity to the recorded cell (Figure 1b). Recordings were carried out continuously at 0.1 Hz stimulation at room temperature. The amplitude of recorded currents varied in the rage 25–450 pA and the currents were completely blocked by bicuculline 10 µM, which confirms their GABAergic nature (Figure 1c).

Two cell populations have been identified based on the sensitivity of GABAergic eIPSCs to Glyght. The first group of cells did not show sensitivity to both conformations of Glyght. On these cells, after 10–15 min of photochrome application, the mean amplitudes of GABAergic eIPSCs were 105.1 ± 6.7% and 108.7 ± 6.6%, respectively, under the action of *trans*-Glyght and after conversion of the compound to the *cis*-configuration using UV irradiation. With the return of the Glyght to the *trans*-configuration by blue light and after washing with control artificial cerebrospinal fluid (ACSF), the mean amplitudes of eIPSCs were 106.7 ± 5.6 and 104 ± 7.4, respectively (Figure 1d–f, *n* = 5; slices from 4 mice: P9–P15).

The second cell population responded on the *trans*-Glyght application (100 µM) by reducing the amplitude of GABAergic eIPSCs to 56.1 ± 8.4% (*p* < 0.05, *n* = 5, 2 mice P12, P13). The inhibitory effect was similar after the conversion of Glyght to the *cis*-configuration (mean amplitude 47.7 ± 7.2% in comparison with control). After the subsequent return to *trans*-configuration, the mean amplitude was even smaller (45.8 ± 10.4%), indicating absence of light-induced modulation and suggesting small continuous run-down of the eIPSCs’ amplitude. Indeed, after washing slices by control ACSF, the amplitude of GABAergic eIPSCs restored, on average, to 75.1 ± 5.93% (*p* < 0.05, *n* = 5, Figure 1g–i).

### 2.2. Analysis of Glyght Action on Paired-Pulse Modulation of GABAergic eIPSCs

To elucidate the reason for the presence of these two different populations of neurons, we first tested the possibility of involving presynaptic mechanisms on the release of neurotransmitters. For this purpose, the analysis of the effect of Glyght on IPSCs induced by paired pulses was carried out.

The paired-pulse modulation (PPM) index was defined as the percentage ratio of eIPSCs’ amplitude evoked by the second (I2) pulse to that evoked by the first (I1) pulse (I2/I1*100). Values of paired-pulse modulation in the control ACSF were taken as 100%. The effect of Glyght on PPM was analyzed separately for the population of cells that responded by a decrease in the amplitude of GABAergic eIPSCs, and for the population that did not show sensitivity to Glyght. In both cases, Glyght did not alter the paired-pulse modulation. Thus, for cells whose amplitudes decreased with the application of Glyght, the PPM was 103.6 ± 12% under the action of *trans*-Glyght and amounted to 109.8 ± 11.3% after switching the compound to the *cis*-configuration (*n* = 5, Figure 2a–c). In the cell population, where we did not observe the effect of the compound on the amplitudes of the first eIPSCs in pairs, the PPM was 107.3 ± 11.5% under the administration of *trans*-Glyght, and 97.6 ± 7.3% after conversion of Glyght to the *cis*-configuration (*n* = 5, Figure 2d–f). These changes were insignificant, within the margin of error, indicating that Glyght does not modulate the mechanisms of neurotransmitter release from presynaptic terminals.

### 2.3. Analysis of Glyght Action on Currents Induced by Extracellular Application of GABA

Then, we evaluated the direct effect of Glyght on postsynaptic GABA_A_ receptors using direct short-term application of GABA (300 µM, for 50 ms) on the surface of neurons. Recordings were carried on CA1 pyramidal neurons of hippocampal slices using the “whole-cell” patch-clamp technique (V_h_ = −70 mV) at room temperature. The arrangement of the application pipette filled with 300 µM GABA and the recording (Rec.) electrode on slices is depicted in Figure 3a. In this case, in addition to synaptically expressed GABA receptors, the agonist also activates extrasynaptic receptors of various subunit composition and properties. In particular, it may be a population of GABA_C_ receptors insensitive to bicuculline [11]. In addition to the high expression of GABA_C_ receptor-forming subunits in the retina and some other parts of the visual pathways, it has also been found in the visual cortex and in the CA1 pyramidal cell layer of the hippocampus [12]. As illustrated in Figure 3b, the recorded GABA-induced transients were only partially inhibited by GABA_A_ antagonist bicuculline. In contrast, picrotoxin, blocker of Cl-selective channels, completely suppressed ionic currents induced by extracellular application of GABA. Strychnine in concentration 1 µM did not affect GABA-induced transients.

Analysis of the effect of Glyght on GABA-induced currents showed that the photochrome does not significantly change their amplitude in either *trans*- or *cis*-configurations. Thus, the mean amplitude of the GABA-induced currents after the addition of *trans*- Glyght was 92.7 ± 1.3% relative to the control. When Glyght was subsequently switched to the *cis*-configuration, the amplitude was 88.1 ± 3.2%; after washing by control ACSF it was 82.6 ± 2.6% (Figure 3c,d).

### 2.4. Involvement of Glycine Receptors in Modulation of the GABAergic eIPSCs in Neurons of Hippocampal Slices

Recent study on the analysis of Glyght action demonstrated that the photochrome effectively inhibits activity of glycine receptors at heterologous expression of different GlyR subunits in cultured cell lines [9]. Glyght (50µM) caused relatively weak inhibition of GlyRs formed by α1 subunits, whereas the effect on α2 GlyRs was much stronger.

Emerging evidence demonstrated the presence of GlyRs in the different areas of the hippocampus [13,14,15], its involvement in tonic regulation of neuronal excitability [16,17,18] and in the modulation of neuronal circuits functioning [19,20,21,22].

Expression of extrasynaptic GlyRs in hippocampal neurons rise the possibility that Glyght might cause effects on GABAergic synaptic currents in brain slices via tonic modulation. To investigate this hypothesis, we analysed firstly the effect of strychnine, antagonist of GlyRs, on the amplitude of the GABAergic eIPSCs and then its effect on preventing inhibitory action of Glyght.

GABAergic eIPSCs were recorded from hippocampal CA1 pyramidal cells using the whole cell patch-clamp technique (V_h_ = 0 mV) at stimulation of every 15 s.

Strychnine replicated the effects of Glyght on the GABAergic eIPSCs: strychnine-resistant and strychnine-sensitive cells were identified in the hippocampal CA1 pyramidal cell layer. Seven out of ten registered cells responded with a significant decrease in the amplitude of eIPSCs to 55.8 ± 8% by the fifth minute of the glycine receptor antagonist administration (Figure 4a–c, *p* < 0.05, paired Student’s *t*-test). The decrease in the amplitude of the currents was reversible: it recovered to 88.2 ± 9.7% after washing (*p* < 0.05, paired Student’s *t*-test). Among the remaining three cells, the mean amplitude of currents after the addition of strychnine was 99.9 ± 4.9% (Figure 4d–f). 

The obtained results suggest that Glyght and strychnine might have the same target when modulating the GABAergic synaptic transmission in CA1 hippocampal region.

### 2.5. The Effect of Glyght on GABAergic Synaptic Transmission Disappeared in the Presence of Strychnine

To verify whether Glyght and strychnine have the same target when modulating the GABAergic synaptic transmission, a series of experiments were carried out in which the photochrome was administered after application of strychnine.

For the strychnine-sensitive neuron illustrated in Figure 5a,b, the amplitude of GABAergic eIPSCs in the control was 169.5 ± 4.3 pA, and it decreased to 136.5 ± 8 pA (*p* < 0.05, two-sample *t*-test) by the fifth minute of strychnine (1 µM) application. The subsequent addition of Glyght (100 µM) in *cis*- and *trans*-configurations did not significantly affect the amplitude of the currents: the amplitudes were 126.4 ± 7.1 pA under *trans*- Glyght and 124.5 ± 6.3 pA under *cis*-Glyght.

On average, amplitude of GABAergic eIPSCs after simultaneous application of *trans*-Glyght and strychnine was 99.2 ± 4.3% relative to the amplitude in the presence of strychnine alone (*n* = 13, Figure 5c).

Thus, Glyght does not affect the amplitude of GABAergic eIPSCs when administered in the presence of strychnine, indicating the same target for both substances in the regulation of GABAergic synaptic transmission.

## 3. Discussion

Glyght is a new photochromic compound which was shown to modulate the activity of glycine receptors in vivo and in vitro models [9]. This compound in *trans*-configuration also caused very weak inhibition of GABA_A_ receptors, which activation results in opening Cl-selective channels. Our present study is focused on determining whether Glyght, which proved to be a photocontrolled negative modulator of postsynaptic glycinergic currents in mouse brain, would affect GABAergic transmission.

Firstly, Glyght was examined on evoked GABAergic postsynaptic currents recorded from the hippocampal CA1 pyramidal cells. We identified two populations of pyramidal cells: some cells were not sensitive to Glyght, whereas others responded with a decrease in the amplitude of GABAergic eIPSCs. The effects were independent of the optical configuration of the compound. Experiments on GABA-induced transients verified that Glyght does not noticeably modulate GABA receptors in mouse hippocampal neurons; however, it obviously affects GABAergic transmission.

Since the photochrome is able to inhibit glycine receptors, and they are present in the hippocampus [21], we tested whether strychnine, glycine receptor antagonist, would act similarly on GABAergic transmission. It turned out that strychnine repeated the effects of Glyght: strychnine-resistant and strychnine-sensitive GABAergic eIPSCs were recorded. After application of strychnine, the photochrome did not affect GABAergic transmission. These data suggest that strychnine and Glyght act on the same target when modulating transmission. This target is likely to be the hippocampal GlyRs, which are expressed extrasynaptically and are involved in the regulation of the excitation/inhibition balance [16,17,18] and modulation of the neuronal circuits functioning [19,20,21,22].

The main reason underlying the presence of two cell populations may be explained by the uneven distribution of the expression of glycine receptors of different subunit composition in the developmental population of cells in the hippocampal CA1 region [19,21]. Analysis of developmental changes in the presence of glycine receptors in hippocampal neurons demonstrated that at postnatal days 1–4 (P1–P4) glycine caused strong elevation of the conductance in neurons, whereas on brain slices from older neonates, it became smaller and negligible at recording on adult brains [23]. This allows us to suggest that the modulating effect may be manifested mainly in young animals.

Glyght modulated synaptic glycine receptors of hypoglossal motoneurons in a photo-controlled manner; however, its effects on GABAergic transmission in the hippocampus were independent of the optical configuration. This may be the result of differences in the subunit composition of extrasynaptically expressed glycine receptors. It may also reflect the features of photochrome isomerization. The use of UV light for switching of the photochrome from *trans* to *cis* configuration is one of the limitations of experimental analysis of Glyght action on pyramidal neurons in brain slices. Prolonged UV irradiation has a phototoxic effect on cells, as the chemical molecules are converted into reactive intermediates that can cause direct local cellular toxicity [24,25]. Additionally, UV radiation can cause polarization of patch-clamp electrodes, which can lead to distortion of electrophysiological signals. For these reasons, it is necessary either to reduce the intensity of UV light, or to avoid direct illumination of the preparation. In our experiments, we applied UV light in the solution outside of the slice to prevent these negative consequences.

On the other hand, decreasing the illumination intensity may lead to only partial isomerization of photochrome molecules, which is not complete even at high light intensity. Indeed, analysis of photochemical properties of Glyght indicate that at high light intensity in DMSO solution photoconversion from *trans* to *cis* isomers is 80% and photoconversion from *cis* to *trans* isomers is 84% Supplementary Table S1 in [9]. In physiological solution, the degree of isomerization can be even less. This suggests that the negligible difference observed in our study between *trans* and *cis* isomers partially rises from this phenomenon of not complete light-induced isomerization of the photochrome.

Another common target for Glyght and strychnine could be nicotine acetylcholine receptors. However, the lack of effects on paired-pulse modulation of GABAergic eIPSCs for both Glyght and strychnine suggests that there was no or minor influence on nicotinic acetylcholine receptors. In addition, the photochrome did not affect muscle nicotinic acetylcholine receptors [9].

Thus, our observations suggest that Glyght does not affect GABA receptors in mouse hippocampal CA1 pyramidal cells, and the inhibition of GABAergic synaptic transmission is mainly associated with modulation of strychnine-sensitive extrasynaptic glycine receptors. Since GABA receptors formed by α1/β2/γ2 subunits are similarly represented in the synaptic and postsynaptic regions of hippocampal and other areas, including cerebral cortex [26,27], one can assume a similarity in the action of Glyght. However, in the future, it will be important to conduct this analysis on the other brain areas.

## 4. Materials and Methods

### 4.1. Animals

Experiments were carried out on laboratory ICR (CD-1) outbred mice of both genders on postnatal days P6-P15. Use of animals was carried out in accordance with the Guide for the Care and Use of Laboratory Animals (NIH Publication No. 85–23, revised 1996) and European Convention for the Protection of Vertebrate Animals used for Experimental and other Scientific Purposes (Council of Europe No. 123; 1985). All animal protocols and experimental procedures were approved by the Local Ethics Committee of Kazan State Medical University (No. 10; 20.12.2016). Mice were kept under natural daylength fluctuations, had free access to food and water and were not involved in any previous procedures.

### 4.2. Brain Slices Preparation

Mice were decapitated, and brains were quickly removed and placed in a Petri dish filled with ice-cold high-K^+^ cutting solution containing 120 mM K-gluconate, 10 mM HEPES-acid, 15 mM Na-gluconate, 0.2 mM EGTA, and 4 mM NaCl (pH 7.2, 290–300 mOsm) for 1 min. The olfactory bulbs, the brainstem with the cerebellum were cut off using a scalpel; the cerebral hemispheres were separated by cutting along the longitudinal fissure and mounted onto the vibratome specimen disc using superglue, orienting them downward with the sagittal cut surface. Sagittal 350–400 μm-thick sections of the cerebral hemispheres containing the hippocampus were prepared in the ice-cold high-K^+^ cutting solution using a tissue slicer of model NVSLM1 (World Precision Instruments, Stevenage, UK). Immediately after cutting, brain slices were incubated for 20 min at room temperature in a bubbled with carbogen gas (a mix of 95% O_2_ and 5% CO_2_) high magnesium artificial cerebrospinal fluid (ACSF) containing 125 mM NaCl, 2.5 mM KCl, 0.8 mM CaCl_2_, 8 mM MgCl_2_, 1.25 mM NaHPO_4_, 14 mM glucose, 25 mM NaHCO_3_ (pH 7.3–7.4, 290–300 mOsm). Before recording, slices were stored for 1–6 h at room temperature in a chamber with carbogen gas bubbled standard ACSF containing 125 mM NaCl, 2.5 mM KCl, 2.3 mM CaCl_2_, 1.3 mM MgCl_2_, 1.25 mM NaHPO_4_, 14 mM glucose, 25 mM NaHCO_3_ (pH 7.3–7.4, 290–300 mOsm).

### 4.3. Electrophysiological Recordings

For the electrophysiological recordings, slices were transferred to a conventional custom-made recording chamber and continuously perfused with standard ACSF saturated with carbogen (the speed of perfusion—25 mL/min). The recordings were carried out at room temperature or at 32 °C. The temperature of 32 °C in the recording chamber was kept by the flow of ACSF through the battery of glass coil condenser just before entering the chamber.

To visualize the slices, an upright microscope Olympus BX51WI (Olympus, Tokyo, Japan) equipped with the iX-on Life 897 EMCCD camera (Andor, Oxford Instruments, Abingdon, UK) and an open source software, Micro-Manager [28] were used. Membrane currents digitized at 10 kHz were recorded at the whole-cell configuration of patch-clamp technique with the holding potentials (V_h_) of −70 mV or 0 mV using an EPC-10 patch clamp amplifier (HEKA Elektronik, Lambrecht (Pfalz), Germany) and PatchMaster v2 × 73.4 software (HEKA Elektronik, Lambrecht, Germany). Patch electrodes (resistance 6–7 MΩ) were filled with the intracellular solution containing 20 mM CsCl, 110 mM CsGluconate, 4 mM MgATP, 10 mM Phosphocreatine, 0.3 mM GTP, 10 mM HEPES, 5 mM EGTA (pH 7.3 with CsOH; 297 mOsm). Membrane GABAergic currents in CA1 pyramidal cells were evoked either by electrical stimulation of interneurons in area CA1 *stratum radiatum* layer or by short-term application of GABA (300 µM, for 50 ms) on a surface of recording cell. For the induction of GABAergic postsynaptic currents, the DS3 Constant Current Isolated Stimulator (Digitimer, Welwyn Garden City, UK) and bipolar glass electrodes were used. The stimulation electrode was placed on the *stratum radiatum* layer in close proximity within the recording cell (distance 70–150 μm). Paired pulses (7–250 µA for 50–200 µs) with intervals of 200 ms were delivered every 10–15 s. To prevent glutamatergic synaptic transmission, CNQX 10 µM and APV 40 µM were added to ACSF. Short-term application of extracellular GABA was carried out via a borosilicate glass micropipette (tip diameter ≈ 50 µm) using VC34 valve-controlled gravity perfusion system (ALA Scientific Instruments, Inc., New York, NY, USA). GABA-induced transients were recorded once per minute in the presence of 1 µM tetrodotoxin and 40 µm D-AP5 in ACSF.

Glyght exists in two optical conformations: *cis* and *trans* (Figure 1a). Switching of the photochrome to the *cis*-configuration is induced by irradiation with UV light, and reverse isomerization by irradiation with blue or visible light. Diodes emitting either UV (365 nm) or blue (455 nm) light and a four-channel LED driver (Thorlabs, Inc., Newton, NJ, USA) were used for these purposes. To prevent direct UV exposure of the slices during switching Glyght, we illuminated solutions in a container, from where it was pumped into a recording chamber.

### 4.4. Drugs

Glyght was used at concentration of 100 µM freshly prepared from 50 mM stock solution dissolved in dimethyl sulfoxide (DMSO). The following stock solutions were prepared using MilliQ water: NMDA receptor antagonist D-AP5 (Hello Bio, Bristol, UK, HB0225)—80 mM, AMPA glutamate receptors antagonist 6-cyano-7-nitroquinoxaline-2,3-dione (CNQX) disodium salt (Hello Bio, Bristol, UK, HB0205)—20 mM, GABA_A_ receptor antagonist (-)-bicuculline methochloride (Tocris, Bristol, UK, 0131)—40 mM, use-dependent Na^+^ channel blocker tetrodotoxin citrate (Hello Bio, HB1035)—2 mM, glycine (Hello Bio, Bristol, UK, HB0299)—100 mM, gamma aminobutyric acid (GABA) (Hello Bio, Bristol, UK, HB0882)—1 M; picrotoxin, 20 mM (Tocris, Bristol, UK, 1128), a GABA_A_ and glycine receptors inhibitor and antagonist of GlyRs, strychnine—1 mM (Sigma-Aldrich, Burlington, MA, USA, 45661) were dissolved in DMSO. All stock solutions were kept at −20 °C.

### 4.5. Data Analysis and Statistics

Ionic currents were visualized and stored using PatchMaster software (HEKA Electronik, Lambrecht, Germany). The initial processing of the results was carried out using Excel worksheets (Microsoft, Redmond, WA, USA); OriginPro 15 software (OriginLab, Northampton, MA, USA) was used to perform a statistical analysis of the data and to plot the graphs. Statistical significance was determined using the paired Student’s t-test with a p-value threshold of 0.05. All data are shown as mean ± SEM.

Igor Pro 6.02 (WaveMetrics, Tigard, OR, USA) and PatchMaster software were used to generate and process superimposed average traces of GABAergic currents.

## Figures and Tables

**Figure 1 ijms-23-10553-f001:**
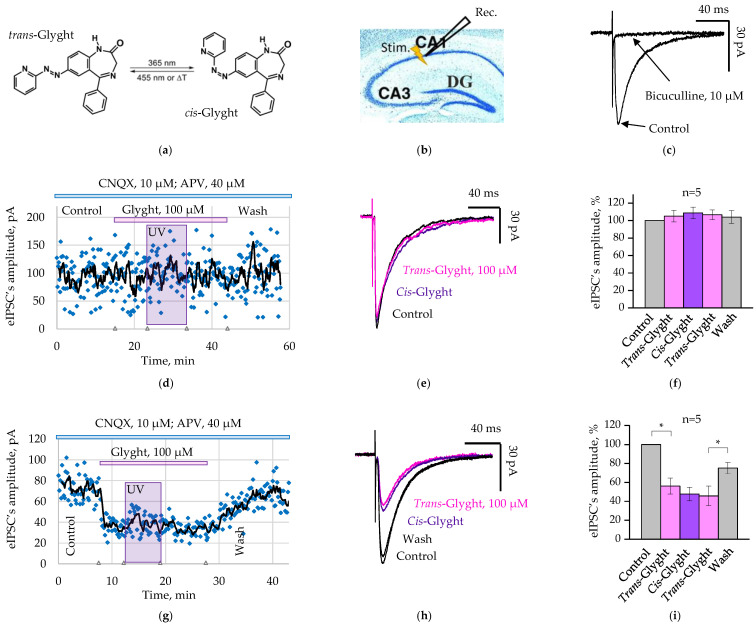
Analysis of Glyght action on the amplitude of the GABAergic eIPSCs in neurons of the hippocampal slices: (**a**) structural formula of Glyght in *trans*- and *cis*-configurations; (**b**) the scheme of location of the bipolar stimulating (Stim.) and the recording (Rec.) electrodes on the hippocampal slice; (**c**) evoked postsynaptic currents in control and in the presence of bicuculline (10 µM). Note that bicuculline completely blocked registered currents, confirming their GABAergic nature; (**d**) original eIPSCs’ amplitude values are continuously displayed on the scatter plot with a 5-period moving-average trendline (black line), obtained from a Glyght resistant neuron; (**e**) example of superimposed traces of eIPSCs in the neuron resistant to Glyght; (**f**) normalized mean amplitudes of eIPSCs in cells resistant to Glyght. The conditions are indicated below the graph. Mean data from 4 mice (P9–P15), *n* = 5; (**g**) Original eIPSCs’ amplitude values are continuously displayed on the scatter plot with a 5-period moving-average trendline (black line), obtained from a Glyght sensitive neuron; (**h**) superimposed examples of traces of eIPSCs in the neuron sensitive to Glyght. Note that the degree of inhibition by Glyght in *trans*- and *cis*-configuration are similar; (**i**) normalized mean amplitudes of eIPSCs in cells sensitive to Glyght. Mean data from two mice (P12, P13), *n* = 5. Data were normalized to control. Mean ± SEM. * Significant difference with *p* < 0.05 (paired Student’s *t*-test).

**Figure 2 ijms-23-10553-f002:**
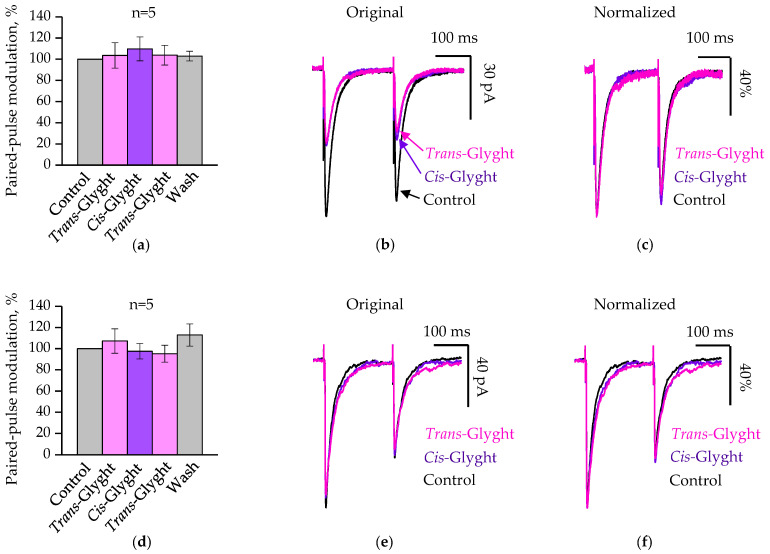
Glyght (100 µM) had no impact on the paired-pulse modulation of GABAergic eIPSCs: (**a**) summary for Glyght-sensitive cells of relative I2/I1 ratio of eIPSCs during application of Glyght in *trans*-, *cis*-configurations and washing. Data are normalized to control. Mean percent ± SEM, *n* = 5, P12, P13; (**b**) superimposed traces of average (*n* = 20) GABAergic eIPSCs, obtained from a Glyght-sensitive cell (mouse P13); (**c**) The superimposed traces scaled to amplitude of the first signal in the pair; (**d**) summary for Glyght-insensitive cells of relative I2/I1 percentage ratio of eIPSCs during application of Glyght in *trans*-, *cis*-configurations and washing. Data are normalized to control. Mean percent ± SEM, *n* = 5, P9–15; (**e**) superimposed traces of average (*n* = 20) GABAergic eIPSCs, obtained from a Glyght-insensitive cell (mouse P12); (**f**) the superimposed traces scaled to amplitude of the first signal in the pair.

**Figure 3 ijms-23-10553-f003:**
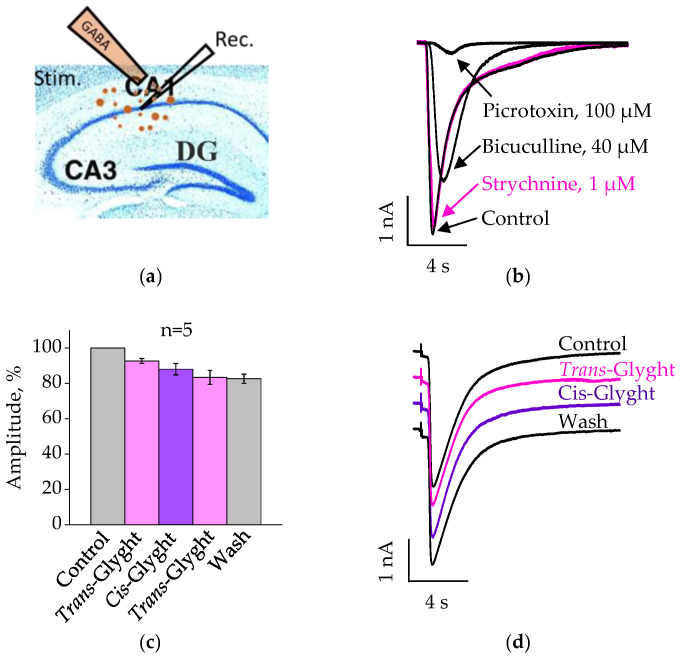
Glyght (100 µM) did not affect the currents induced by extracellular application of GABA on the hippocampal CA1 pyramidal neurons: (**a**) the position of the application pipette filled with 300 µM GABA and the recording electrode (Rec.) on a slice; (**b**) superimposed traces of average (*n* = 4–8) GABA-induced currents recorded in control ACSF, and after addition of 1 µM of strychnine, 40 µM of bicuculline and 100 µM picrotoxin (mouse P8). Note complete inhibition of the GABA-induced currents by picrotoxin; (**c**) average normalized amplitudes of GABA-induced currents after application of Glyght in *trans*-, *cis*-configurations and after washing as a percentage of the controls. Mean percent ± SEM, *n* = 5, P7–8; (**d**) superimposed examples of traces of average (*n* = 5–7) GABA-induced currents recorded in control ACSF, after application of Glyght in *trans*-, *cis*-configurations and washing. Recordings were obtained from P7 mouse.

**Figure 4 ijms-23-10553-f004:**
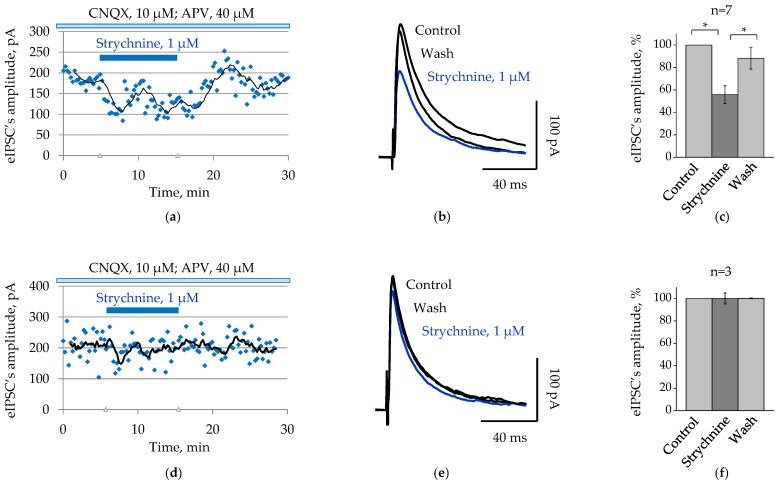
Strychnine-sensitive and strychnine-resistant GABAergic eIPSCs were recorded in the hippocampal CA1 pyramidal cell layer: (**a**,**d**) the original eIPSCs’ amplitude values are continuously displayed on the scatter plot with a 5-period moving-average trendline (black line): the first cell responded with a reversible decrease in the amplitude of GABAergic eIPSCs to the administration of strychnine (1 µM), whereas in the second cell strychnine did not affect the GABAergic eIPSCs; (**b**,**e**) superimposed traces of average GABAergic eIPSCs (*n* = 20–27) recorded in control ACSF, after application of strychnine and washing (mouse P6); (**c**,**f**) normalized to control values mean amplitudes of GABAergic eIPSCs after application of strychnine (1 µM) and subsequent washing for strychnine-sensitive (**c**) and strychnine-resistant (**f**) neurons (mean percent ± SEM; P6–15). * Significant difference with *p* < 0.05 (paired Student’s *t*-test).

**Figure 5 ijms-23-10553-f005:**
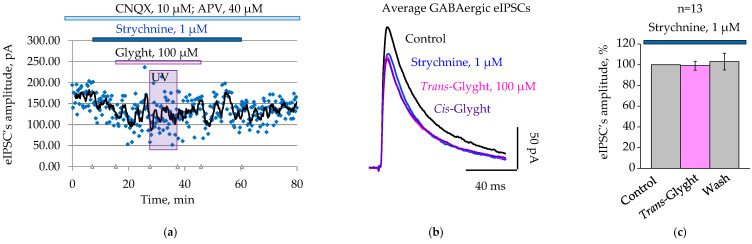
Glyght (100 µM) did not affect GABAergic transmission after application of strychnine: (**a**) the original eIPSCs’ amplitude values are continuously displayed on the scatter plot with a 5-period moving-average trendline (black line): note the absence of an additional decrease in amplitude of GABAergic eIPSCs after administration of Glyght in both *cis*- and *trans*-configurations; (**b**) superimposed traces of average GABAergic eIPSCs (*n* = 9–20) recorded in control ACSF, after application of strychnine, Glyght and after washing; (**c**) summary results of the normalized mean amplitudes of eIPSCs after application of Glyght (100 µM) and subsequent washing, whereas strychnine was routinely added to solutions (mean percent ± SEM; *n* = 13; P6–P8).

## Data Availability

https://drive.google.com/drive/folders/17eJ_TW4-bkxs9-Pg1yMeo_PA1xodKgEg?usp=sharing (accessed on 16 August 2022).

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
