# Peer review of "Action of the Photochrome Glyght on GABAergic Synaptic Transmission in Mouse Brain Slices"

_ijms, 2022, doi:10.3390/ijms231810553_

Round 1

Reviewer 1 Report

In this manuscript the authors investigated the effect of the new photochromic compound of Glyght on GABAergic transmission in the hippocampus of mice following several other photochromes were developed by the group. Whole-cell patch-clamp recording was carried out to determine the Glyght action on evoked GABAergic inhibitory postsynaptic currents, and two populations of cells responding to Glyght action were identified. Outcomes suggested that modulating GABAergic synaptic transmission via action on extrasynaptic glycine receptors is presumably the mechanism of Glyght’s effectiveness. The purpose of this research is interesting, however most of the data was already shown in previous studies, so more detailed mechanism is essential to be published in International Journal of Molecular Sciences.

-          The authors for the first time investigated the Glyght action on evoked GABAergic inhibitory postsynaptic currents in hippocampus of mouse, and two populations of pyramidal neurons responding to Glyght were identified. However, not sure if the authors investigated the cortices of mouse brain, and whether those outcomes are consistent or not.

-          Limitation of this study is missing;

-          The discrimination of the two types of pyramidal neurons in CA1 of hippocampus responding to Glyght raised up essential question of the mechanism. More detailed discussion for this would be desired;

Author Response

Replies to referees.

We are grateful to reviewers for careful analysis of our manuscript.

The paper was revised according to the reviewer’s suggestion and below are point- by-point replies.

Referee 1.

  1. The authors for the first time investigated the Glyght action on evoked GABAergic inhibitory postsynaptic currents in hippocampus of mouse, and two populations of pyramidal neurons responding to Glyght were identified. However, not sure if the authors investigated the cortices of mouse brain, and whether those outcomes are consistent or not.

Reply:

We performed analysis of Glyght only on hippocampal brain slices. Since GABA receptors formed by α1/β2/γ2 subunits are similarly represented in the synaptic and postsynaptic regions of hippocampal and cortical neurons (Fritschy and Brünig, 2003; Mohler, 2006), one can assume a similarity in the action of Glyght. However, in the future it will be important to conduct this analysis on the cerebral cortex.

We added corresponding text to the manuscript.

References:

Fritschy JM, Brünig I (2003) Formation and plasticity of GABAergic synapses: physiological mechanisms and pathophysiological implications. Pharmacol Ther 98:299–323

Möhler, H., 2006. GABAA receptor diversity and pharmacology. Cell and tissue research326(2), pp.505-516.

  1. Limitation of this study is missing;

Reply:

The use of UV light for switching of the photochrome from trans to cis configuration is one of limitations of experimental analysis of Glyght action on pyramidal neurons in brain slices. Prolonged UV irradiation has a phototoxic effect on cells, as the chemical molecules are converted into reactive intermediates that can cause direct local cellular toxicity (Laissue et al., 2017; Marek et al., 2019). Also, UV radiation can cause polarization of patch-clamp electrodes, which can lead to distortion of electrophysiological signals. For these reasons, it is necessary either to reduce the intensity of UV light, or to avoid direct illumination of the preparation.  In our experiments, we applied UV light in the solution outside of the slice to prevent these negative consequences.

On the other hand, decreasing the illumination intensity may lead to only partial isomerisation of photochtome molecules, which is not complete even at high light intensity. Indeed, analysis of photochemical properties of Glyght indicate that at high light intensity in DMSO solution photoconversion from trans to cis isomers is 80% and photoconversion from cis to trans isomers is 84% (Gomila et al., 2020, Supplementary Table1). In physiological solution, the degree of isomerisation can be even less. This suggests that the observed in our study negligible difference between trans and cis isomers partially rises from this phenomenon of not complete light-induced isomerisation of the photochrome. 

We added corresponding text in the discussion.

References:

Marek, V., Potey, A., Réaux-Le-Goazigo, A., Reboussin, E., Charbonnier, A., Villette, T., Baudouin, C., Rostène, W., Denoyer, A. and Parsadaniantz, S.M., 2019. Blue light exposure in vitro causes toxicity to trigeminal neurons and glia through increased superoxide and hydrogen peroxide generation. Free Radical Biology and Medicine131, pp.27-39.

Laissue, P.P., Alghamdi, R.A., Tomancak, P., Reynaud, E.G. and Shroff, H., 2017. Assessing phototoxicity in live fluorescence imaging. Nature methods14(7), pp.657-661.

Gomila, A.M., Rustler, K., Maleeva, G., Nin-Hill, A., Wutz, D., Bautista-Barrufet, A., Rovira, X., Bosch, M., Mukhametova, E., Petukhova, E. and Ponomareva, D., 2020. Photocontrol of endogenous glycine receptors in vivo. Cell chemical biology27(11), pp.1425-1433.

3      The discrimination of the two types of pyramidal neurons in CA1 of hippocampus responding to Glyght raised up essential question of the mechanism. More detailed discussion for this would be desired;

Reply:

Two types of GABAergic eIPSCs were detected in our study: Glyght sensitive, and Glyght insensitive. This correlated with presence of strychnine-sensitive and strychnine-resistant GABAergic eIPSCs recorded in the hippocampal CA1 brain slices. The main reason underlying the presence of these two populations may be the difference in the expression level of non-synaptic glycine receptors.

Developmental changes in the presence of glycine receptors in hippocampal neurones was earlier described. At postnatal days 1-4 (P1-P4) glycine caused strong elevation of the conductance in neurons, while on brain slices from older neonates it become smaller and negligible at recording on adult brains (Ito and Cherubini, 1991). This allows us to suggest that the modulating effect may be manifested mainly on young animals

The corresponding text is added to the discussion.

Reviewer 2 Report

The study explores common target for the benzodiazepine derivative photochrome Glyght and strychnine. Data suggest that Glyght does not affect either GABAa or nicotinic acetylcholine receptors in mouse hippocampal CA1 pyramidal cells. It is suggested, that the inhibition of GABAergic synaptic transmission by Glyght is mainly associated with the modulation of strychnine-sensitive extrasynaptic glycine receptors. I would accept the claim, however several data need clarification. These are:

1)      variable effects of GABAa receptor antagonist bicuculline: cf. Figure 1C versus Figure 3b;

2)      effect of picrotoxin higher than that of bicuculline: cf. Figure 3b;

3)      negligible different between trans and cis isomers;

4)      gentle strychnine-sensitivity (Figure 4b).

Authors conjecture the involvement of extrasynaptic GlyRs in the regulation of the excitation/inhibition balance and modulation the neuronal circuits functioning via Glycinergic tonic inhibition. Interesting idea, however call for a more detailed elucidation.

Author Response

Replies to referees.

We are grateful to reviewers for careful analysis of our manuscript.

The paper was revised according to the reviewer’s suggestion and below are point- by-point replies.

Referee 2.

1)      variable effects of GABAa receptor antagonist bicuculline: cf. Figure 1C versus Figure 3b;

Reply:

Difference in the efficacy of bicuculline action presented in these two figures results from the way of generation of GABA-induced responses. In the Figure 1C are presented results of recording of synaptic currents induced by release of GABA from presynaptic terminals and activating population of bicuculline sensitive GABAA receptors formed by α1/β2/γ2 subunits in CA1 hippocampal neurons.

The Figure 3b illustrates recording of responses induced by GABA released from the application pipette. In this case the agonist activates, in addition to synaptically expressed GABAA receptors, also extrasynaptic receptors of different subunit composition and properties. In particular, it may be the population of bicuculline-insensitive GABAC receptors (Polenzani et al., 1991). Indeed, in addition to high expression of subunits forming GABAC receptors in retina and some other parts of the visual pathways, it was also detected in the visual cortex and in the CA1 pyramidal cell layer of hippocampus (Wegelius et al.,1998).

In contrast, picrotoxin blocks Cl-selective channels and is capable of suppressing ionic currents caused by activation of both GABAA and GABAC receptors.

The corresponded text is added to “Results” section.

References

  Polenzani, L., Woodward, R.M. and Miledi, R., 1991. Expression of mammalian gamma-aminobutyric acid receptors with distinct pharmacology in Xenopus oocytes. Proceedings of the National Academy of Sciences88(10), pp.4318-4322.

Wegelius, K., Pasternack, M., Hiltunen, J.O., Rivera, C., Kaila, K., Saarma, M. and Reeben, M., 1998. Distribution of GABA receptor ρ subunit transcripts in the rat brain. European Journal of Neuroscience10(1), pp.350-357.

2)      effect of picrotoxin higher than that of bicuculline: cf. Figure 3b;

Reply:

We suggest that higher effect of picrotoxin than that of bicuculline observed in our experiments results from:

(i) difference in the mechanism of compounds action;

(ii) differences in the used concentrations of compounds;

(iii) presence of bicuculline-insensitive GABAC receptors. 

Bicuculline acts as a competitive antagonist at GABAA receptors, while picrotoxin is a non-competitive antagonist that blocks chloride-selective channels activated by GABA.  Hippocampal and cortical neurons express GABAA receptors formed by α1/β2/γ2 subunits (Möhler, 2006). Cl-selective Ion channel formed by these subunits had an IC50 for picrotoxin of IC10 = 1.3 µM (Curley et al., 1995; Zhorov and Bregestovski, 2000), i.e. its effect at used in our experiments concentration should be strong.

It has been shown that at application of equal concentration picrotoxin is more efficient than bicuculline (Huang et al., 2003). In the experiment presented on the fig.3 we used 100µM of picrotoxin and 40 µM bicuculline.

In addition, presence of bicuculline-insensitive GABAC receptors (see reply to the question 1).

All this should explain the difference in the inhibitory effectiveness of the compounds.

References:

Möhler, H., 2006. GABAA receptor diversity and pharmacology. Cell and tissue research326(2), pp.505-516.Gurley, D., Amin, J., Ross, P.C., Weiss, D.S. and White, G., 1995. Point mutations in the M2 region of the alpha, beta, or gamma subunit of the GABAA channel that abolish block by picrotoxin. Receptors & channels3(1), pp.13-20.

Gurley, D., Amin, J., Ross, P.C., Weiss, D.S. and White, G., 1995. Point mutations in the M2 region of the alpha, beta, or gamma subunit of the GABAA channel that abolish block by picrotoxin. Receptors & channels3(1), pp.13-20.

Zhorov, B.S. and Bregestovski, P.D., 2000. Chloride channels of glycine and GABA receptors with blockers: Monte Carlo minimization and structure-activity relationships. Biophysical Journal78(4), pp.1786-1803.

Huang, S.H., Duke, R.K., Chebib, M., Sasaki, K., Wada, K. and Johnston, G.A., 2003. Bilobalide, a sesquiterpene trilactone from Ginkgo biloba, is an antagonist at recombinant α1β2γ2L GABAA receptors. European journal of pharmacology464(1), pp.1-8.

3)      negligible different between trans and cis isomers;

Reply:

Two main reasons are suggested for explaining this negligible difference.

  1. In hippocampal neurons are expressed GlyRs formed by alpha3 subunits, which might be less sensitive to action of different isomers of Glyght. We have not analysed properties of this subunit at heterologous expression.

2.Presence of mixed trans/cis Glyght isomers. Indeed, analysis of photochemical properties of Glyght indicate that in DMSO solution photoconversion from trans to cis isomers is 80% and photoconversion from cis to trans isomers is 84% (Gomila et al., 2020, Supplementary). In physiological solutions, the degree of isomerisation can be even less. This suggests that the observed in our study negligible difference between trans and cis isomers partially rises from this phenomenon of not complete light-induced isomerisation.

Some points are added to the text.

Reference:

Gomila, A.M., Rustler, K., Maleeva, G., Nin-Hill, A., Wutz, D., Bautista-Barrufet, A., Rovira, X., Bosch, M., Mukhametova, E., Petukhova, E. and Ponomareva, D., 2020. Photocontrol of endogenous glycine receptors in vivo. Cell chemical biology27(11), pp.1425-1433.

4)      gentle strychnine-sensitivity (Figure 4b).

Reply:

Gentle strychnine-sensitivity is not surprising, as on figures 4 and 5 are shown GABAergic eIPSCs. Glycine receptors just modify the amplitude of these GABAergic responses.
